# Dolutegravir: Virologic response and tolerability of initial antiretroviral regimens for adults living with HIV

**Analú Correa[1], Polyana Monteiro[1‡], Fernanda Calixto[2‡], Joanna d'Arc Lyra Batista[3‡], Ricardo Arraes de Alencar Ximenes[1,4], Ulisses Ramos Montarroyos[5]\***

1 Medical Sciences College, University of Pernambuco, Recife, Pernambuco, Brazil, 2 Teaching Hospital Oswaldo Cruz, University of Pernambuco, Recife, Pernambuco, Brazil, 3 Medical Sciences College, Federal University of Fronteira Sul, Chapecó, Santa Catarina, Brazil, 4 Post-Graduation Program in Tropical Medicine, Federal University of Pernambuco, Recife, Pernambuco, Brazil, 5 Institute of Biological Sciences, University of Pernambuco, Recife, Pernambuco, Brazil

๏ These authors contributed equally to this work.
‡ These authors also contributed equally to this work.
\* ulisses.montarroyos@upe.br

**Data Availability Statement:** The data underlying the results presented in the study are available as Supporting Information files.

## Abstract

The integrase inhibitor dolutegravir was included in initial antiretroviral therapy in Brazil in January 2017. Studies have demonstrated that the efficacy and safety of antiretrovirals have improved with the introduction of new classes of antiretrovirals, such as integrase inhibitors. This study aimed to estimate the frequency of individuals with a virologic response by week 24 of antiretroviral treatment and to describe the adverse events of the regimen containing dolutegravir. This was a cohort of people living with HIV followed up at a referral hospital. Patients were included who had initiated their first treatment between January and August 2017. Data were obtained from medical records, the Drug Logistics Management System and from the Laboratory Tests Control System. Two hundred and twenty-two patients were included for the tolerability analysis and one hundred and thirty-seven for the virologic response analysis. The mean age was 34 years, the median time between diagnosis and initiating treatment was 1.9 months and the median time on antiretroviral therapy was 13.2 months. The frequency of adverse events was 10% (95% CI: 7% to 15.2%), of these, amongst the most frequent events, 91% presented gastrointestinal effects, and 47.8% neuropsychiatric. By week 24 the estimated incidence of virologic response was 89.1% (95% CI: 83% to 93.5%), with an increase during the first 6 months in the number of T-CD4 lymphocytes of 50.7 cells/mm 3 (95% CI: 42 to 59.3). Initial antiretroviral regimens containing dolutegravir were well tolerated and effective in viral suppression during the first 24 weeks after initiating treatment. The occurrence of adverse events was low, either mild or moderate.

**Funding:** This study was financed by the Coordenação de Aperfeiçoamento de Pessoal de Nível Superior - Brasil (CAPES) - Finance code 001 and Fundação de Amparo à Ciência e Tecnologia do Estado de Pernambuco - FACEPE (State of Pernambuco Science and Technology Support Foundation – process IBPG-1420-4.01/16) to URM. RAAX was supported by CNPq (Scholarship 308311/2009-4). The funders had no role in study design, data collection and analysis, decision to publish, or preparation of the manuscript.

**Competing interests:** The authors have declared that no competing interests exist.

## Introduction

Antiretroviral therapy transformed the natural history of HIV infection [1,2]. Advances in treatment have severely impacted the course of the disease, reducing morbidity and mortality and improving the quality of life of people living with HIV (PLHIV). However, there are adverse events associated with antiretroviral use as well as other factors that contribute to poor adherence and to the discontinuation of treatment and may lead to therapeutic failure and an accumulation of mutations [3].

Dolutegravir (DTG) is an integrase inhibitor that was incorporated into the Brazilian pharmacological arsenal as of January 2017. It has become the third preferred agent to compose the initial antiretroviral regimen of HIV-1 positive individuals, according to Brazil's treatment guidelines [4]. It is administered once a day in a single 50mg tablet, without the need for pharmacological reinforcement and with the advantage of presenting few drug-drug interactions [5,6].

Controlled, randomized trials have compared the efficacy of DTG in treatment-naïve individuals with drugs commonly used as a choice in major treatment guidelines. In the Single study [7], the comparison was made with efavirenz (EFZ), in the Spring-2 study [8] with raltegravir (RAL) and in the Flamingo study [9] with darunavir These studies verified that dolutegravir provided non-inferior efficacy compared to raltegravir, and was superior in comparison to darunavir, and also provided superior virologic efficacy compared to the regimen using efavirenz [7–9].

In relation to tolerability, the Spring-2 study reported similar adverse event results when compared to the use of integrase inhibitors as a third drug. In the Flamingo study [9], there was a report 1 life threatening event related to DTG (a suicide attempt), however the patient had a history of depression. The Single study [7] reported fewer adverse events using DTG (10% versus 16%) despite more frequent reports of insomnia, which were more intense than in previous studies.

Rates of treatment discontinuation were lower in the group using DTG in the three abovementioned studies.

In a non-controlled study in Northern Ireland in 2017, Todd et al., in a cohort of with 51 treatment-naive individuals and 106 treatment-experienced individuals, reported a rapid reduction in the viral load in the treatment-naive group, with 94% presenting an undetectable viral load and a 30% improvement in the mean CD4 cell count at week 12 of treatment. In terms of tolerability, three patients discontinued DTG due to events such as insomnia, depressed mood, and anxiety [10].

In 2019, a Brazilian cohort study observed that more than 90% of the patients treated with DTG achieved viral suppression. The adverse event wasn't studied. The study concludes that there is evidence of the superiority of dolutegravir over efavirenz, lopinavir and atazanavir in suppressing viral replication in adults by the first year on ART, based in a national approach [11]. Brazil have different social and economic aspect between the regions of country and within of each region, because this is important that local studies to be carried and the evidence compared with the national reality.

The present study aims to estimate the frequency of individuals with a virologic response up until the 24th week after initiating antiretroviral treatment, and to describe the occurrence of adverse events of the initial DTG regimen in people living with HIV/AIDS treated in a hospital of northeast of Brazil.

## Materials and methods

### Study design and setting

This was a retrospective cohort study conducted at a referral hospital for HIV/AIDS, Hospital Correia Picanço, located in the city Recife, Pernambuco, Brazil. Data collection took place

from April to October 2018, the inclusion criteria of study population consisted of individuals with HIV infection aged 18 years or older, who had initiated the first antiretroviral regimen containing dolutegravir between January and August 2017, according to Brazilian Ministry of Health guidelines [4].

## Data collection

The initiation date of the antiretroviral regimen was considered as the starting moment of the cohort. Measurements of the CD4 and viral load and the occurrence of adverse events were observed within a minimum of 24 weeks after initiating the regimen. For those with no results of viral load and CD4 cell count up until 24 weeks, we considered the measurements up until 36 weeks. Were excluded the patients who had undetected viral load (<50 copies/ml) at baseline and who don't have at least two available HIV viral load measurements collected at different timepoint. For standardization purposes, all viral load and CD4 records of the study participants were collected following the hospital routine. Due to this, is expected not all patients had viral load and CD4 cell count measurements consistently.

Eligible patients were identified using the Drug Logistics Management System (known as SICLOM). T-CD4 lymphocyte and viral load measurements were collected from the Laboratory Tests Control System (SISCEL). SICLOM and SISCEL are integrated computerized information systems for monitoring PVHIV, under the administration of the Brazilian Ministry of Health, which stores data on laboratory testing, drug dispensing and stocks. Information related to biological and behavioural characteristics, comorbidities, initial regimen, concomitant use of other medications, regimen change, and adverse events were collected from patient records.

The collected information was recorded on a specific research form, corrected for possible inconsistencies, and after a review, the forms were submitted for typing.

Our primary endpoint was the proportion of participants with viral load of less than 50 copies per mL at 24th week of retroviral treatment [4] and proportion of presence of adverse events in the same period. The adverse events were classified in gastrointestinal symptoms, neuropsychiatric and other symptoms [12]. The secondary endpoint was to describe the variations of CD4 counts over time in patients living with HIV.

## Statistical methods

The frequency of therapeutic response was represented by the cumulative incidence measure and estimated with their respective 95% confidence intervals. The characteristics of the study population were presented by frequency distribution when the variables were categorical and by means and standard deviation for quantitative variables. The hypothesis of normal distribution was observed with the Shapiro-Wilk test, presenting the median and interquartile range when the hypothesis of normality was rejected. In the analysis of the association of the explanatory factors of the virological response, Pearson's chi-square test and Student's t test or Kruskal-Wallis test were used to compare means or medians, respectively. To estimate the mean growth rate in the number of T-CD4 cells over time, a Generalized Estimating Equation (GEE) model was used for repeated measurements. The significance adopted in the study was 5% ($p < 0.05$). The software used for the analysis was Stata 14.

## Ethics statement

The study was approved by the Teaching Hospital Oswaldo Cruz of University of Pernambuco (HUOC) Research Ethics Committee in accordance with resolution 466/12, under Report No. 2,600,137, on 16/04/2018.

## Results

Of the 562 individuals taking dolutegravir registered on SICLOM, we identified 230 who initiated the first antiretroviral regimen containing dolutegravir (Fig 1).

A total of 222 individuals were included in the tolerability analysis, of whom eight were excluded as losses during the collection period: one had no detailed information regarding the regimen used, four had no records of the initiation date of antiretroviral therapy (ART) and three had no information regarding regimen change and the occurrence of adverse events. For the virological response analysis, 93 individuals provided did not have laboratory data, thereby leaving 137 individuals to compose the sample.

The study population consisted of 72.5% male individuals with a mean age of 34.5 years. With regard to habits, 17.3% of the respondents used illicit drugs, 64.8% consumed alcohol and 27.6% were smokers.

Of the total patients, 25.8% regularly used another medication, 0.5% presented with diabetes mellitus, 5.4% had been diagnosed with systemic arterial hypertension and 10.3% other diseases.

A total of 220 participants (99.1%) had initiated a combination of dolutegravir, tenofovir and lamivudine (DTG + TDF + 3TC). The median time between diagnosing the disease and initiating treatment was 1.9 months and the median time of using ART was 13.2 months. The median maximum viral load was 33,400 copies/mm$^3$ and the CD4 nadir median was 360 cells/mm$^3$. These characteristics are presented in Table 1.

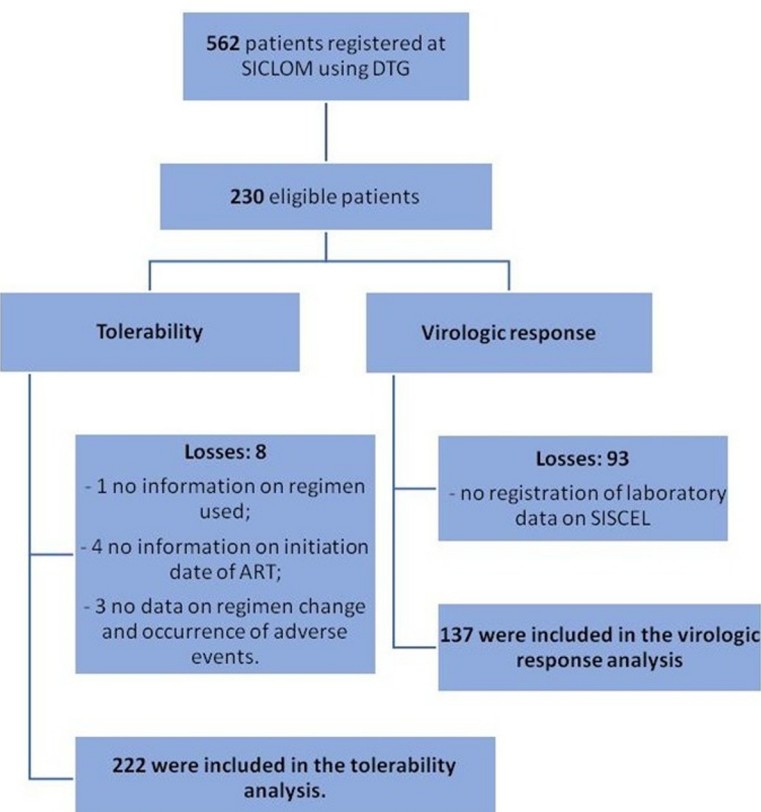

**Fig 1. Study flowchart.**

**Table 1. Biological, clinical and behavioural characteristics of 222 people living with HIV who initiated the first antiretroviral regimen containing dolutegravir.**

| Characteristics | Number/Total (%) |
|---|---|
| **Biological** | |
| Age[a] | 34.5 ± 9.7 (17.4–66.5) |
| Sex: Male | 161/222 (72.5) |
| **Habits** | |
| Use of illicit drugs | 161/222 (17.3) |
| Alcohol consumption | 94/145 (64.8) |
| Smoker | 39/141 (27.6) |
| **Clinical characteristics related to HIV** | |
| Regular use of other medication | 48/186 (25.8) |
| Comorbidities | |
| No comorbidities | 186/222 (83.8%) |
| Diabetes | 1/222 (0.5%) |
| Arterial hypertension | 12/222 (5.4%) |
| Others | 23/222 (10.3%) |
| Time between HIV diagnosis and initiating ART (months)[b] | 1.9 (24 days–3.4 months) |
| Time on ART (months)[b] | 13.2 (12.7–14 months) |
| ART regimen | |
| DTG+TDF+3TC | 220/222 (99.1%) |
| DTG+ABC+3TC | 2/222 (0.9%) |
| Max. Viral load (copies/mm3)[b] | 33.400 (6.934–186.352) |
| CD4 nadir (cell/mm$^3$)[b] | 360 (167–549) |

[a] Mean ± sd (Minimum—Maximum).

[b] Median (P25—P75). HIV—human immunodeficiency virus. ART—antiretroviral therapy. DTG–dolutegravir. TDF—tenofovir. 3TC—lamivudine. ABC–abacavir

The frequency of adverse events following the introduction of ART was 10.4% (95% CI: 7% to 15.2%). Of the 23 individuals that presented adverse events, gastrointestinal symptoms were the most frequent, representing 91% (21/23) of patient complaints. Diarrhea, nausea, vomiting, heartburn and stomach pains as the main adverse events reported. Neuropsychiatric symptoms were reported by 47.8% (11/23) of individuals, with insomnia and dizziness being the main complaints. Amongst other adverse events (34.8%), one patient reported fever and other patients reported skin blemishes, cough and itching.

Most of those who used this regimen and reported adverse events only reported one single complaint (43.5%), while 34.8% reported at least two complaints and 21.7% three or more complaints.

Amongst those who needed to change their ART (4.0%), the most frequent reason was tuberculosis—HIV coinfection. Three other cases of change occurred, one due to pregnancy, an individual with a psychiatric disease prior to initiating ART, and one for reasons unregistered in the medical records (Table 2).

The estimated incidence of virologic response by week 24 of treatment was 89.1% (95% CI: 83% to 93.5%). The median viral load of patients at the beginning of ART was 334,000 copies, after 24 weeks decrease to 193 copies. Analysing the association of virologic response and the factors studied, in the block of variables of biological and behavioural characteristics, none demonstrated an association with virologic response. Amongst the clinical variables, a statistically significant association between the maximal viral load and virologic response was

**Table 2. Adverse events after introducing the first antiretroviral regimen containing dolutegravir.**

| Adverse events | Number (%) |
|---|---|
| **Adverse event** | |
| Yes | 23 (10.4%) |
| No | 199 (89.6%) |
| **Adverse events presented[a]** | |
| **Gastrointestinal symptoms** | 21 (91%) |
| Diarrhea | 10 (43.4%) |
| Nausea | 4 (17.3%) |
| Vomits | 3 (13.0%) |
| Heartburn/stomach pains | 4 (17.3%) |
| **Neuropsychiatric symptoms** | 11 (47.8%) |
| Insomnia | 3 (13.0%) |
| Dizziness | 2 (8.7%) |
| Headaches | 1 (4.3%) |
| Bilateral visual changes | 1 (4.3%) |
| Anxiety/Distress | 1 (4.3%) |
| Dormancy | 1 (4.3%) |
| Forgetfulness | 1 (4.3%) |
| Irritability | 1 (4.3%) |
| **Others** | 8 (34.8%) |
| **Number of adverse events presented** | |
| One | 10 (43.5%) |
| Two | 8 (34.8%) |
| Three or more | 5 (21.7%) |
| **Change of regimen** | |
| No | 213 (95.9%) |
| Yes | 09 (4.1%) |
| **Reason for change** | |
| TB treatment | 6 (66.7%) |
| Pregnancy | 1 (11.1%) |
| Psychiatric outbreak | 1 (11.1%) |
| Reason not specified | 1 (11.1%) |

[a] Non-concomitant categories

observed, with a higher viral load amongst those who presented no virologic response up until week 24, as well as a significant association between CD4 nadir, with lower counts in the group with no virologic response, as well as a significant association between the CD4 nadir with the lowest count in the group with no virologic response up until week 24 (Table 3).

With regard to the immune response, the median CD4 cell count at baseline was 346 cells/mm$^3$, presenting a monthly growth rate of 25.0 cells/mm$^3$ (95% CI: 19.6 to 30.3). During the first 6 months, the increase mean was 50.7 cells/mm$^3$ (95% CI: 42 to 59.3) per month. The reference during first 12 months was the average increase mean was 37.7 cells/mm$^3$ (95% CI: 31.7 to 43.8) per month (Fig 2).

## Discussion

In the present study we observed that dolutegravir is an efficient, well tolerated drug, not only in clinical trials, but also in clinical practice where other factors could influence the therapeutic

**Table 3.** Analysis of the association of covariates with the virologic response at week 24 in individuals who initiated the first antiretroviral regimen containing dolutegravir.

| Variables | Virologic response (n = 122) | No virologic response (n = 15) | p-value |
|---|---|---|---|
| **Biological aspects** | | | |
| **Age**[a] | 34.6 ± 9.8 | 36.8 ± 10.6 | 0.419 |
| **Sex** | | | 0.623 |
| Female | 41 (33.6%) | 6 (40.0%) | |
| Male | 81 (66.4%) | 9 (60.0%) | |
| **Habits** | | | |
| **Use of illicit drugs** | | | |
| Yes | 18 (18.6%) | 0 (0%) | 0.207[d] |
| No | 79 (81.4%) | 10 (100%) | |
| Not informed | 25 | 5 | |
| **Alcohol consumption** | | | |
| Yes | 61 (64.9%) | 6 (60.0%) | 0.741[d] |
| No | 33 (35.1%) | 4 (40.0%) | |
| Not informed | 28 | 5 | |
| **Smoker** | | | |
| Yes | 28 (31.8%) | 2 (22.2%) | 0.716[d] |
| No | 60 (68.2%) | 7 (77.8%) | |
| Not informed | 34 | 6 | |
| **Clinical characteristics related to HIV** | | | |
| **Regular use of other medication** | | | |
| Ye | 27 (25.0%) | 4 (26.7%) | 0.889[d] |
| No | 81 (75.0%) | 11 (73.3%) | |
| Not informed | 14 | 0 | |
| **Comorbidity** | | | |
| Yes | 96 (78.7%) | 12 (80.0%) | 0.907[d] |
| No | 26 (21.3%) | 3 (20.0%) | |
| **Time between HIV diagnosis and initiating ART (months)**[b] | 1.93 (1.1–3.9) | 1.87 (0.4–2.4) | 0.415 |
| **Time on ART (years)**[b] | 1.07 (1.03–1.11) | 1.08 (1.03–1.14) | 0.416 |
| **ART Regimen** | | | |
| DTG+TDF+3TC | 121 (99.2%) | 15 (100%) | 1.000 |
| DTG+ABC+3TC | 1 (0.8%) | 0 (0%) | |
| **Maximum viral load S(x$10^{-3}$; copies/mm$^3$)**[b] | 33.0 (7.1–86.7) | 446.7 (142.2–943) | 0.002[c] |
| **CD4 nadir (cell/mm$^3$)**[b] | 392 (191–584) | 166 (77–323) | 0.027[c] |

[a] Mean ± sd–t student test

[b] Median (P25—P75)—Mann-Whitney test

[c] Statistically significant difference

[d] Fischer exact test HIV—human immunodeficiency virus. ART—antiretroviral therapy. DTG–dolutegravir. TDF—tenofovir. 3TC–lamivudine. ABC–abacavir.

response. The study has demonstrated a high percentage of individuals with an undetectable viral load after 24 weeks of antiretroviral therapy (ART) initiation with a rapid virologic response. These results are similar to other previous studies [7,9,10].

The results presented demonstrated that the estimated incidence of response up until the 24th week of treatment with antiretroviral regimen containing dolutegravir was 89.1%. In addition to virologic suppression, during the first six months, the individuals presented an increase in the number of T-CD4 lymphocytes of 50.7 cells/mm$^3$ (95% CI: 42 to 59.3). A recent Brazilian cohort published in 2019 estimated a viral suppression of 81.8% in 18000 participants

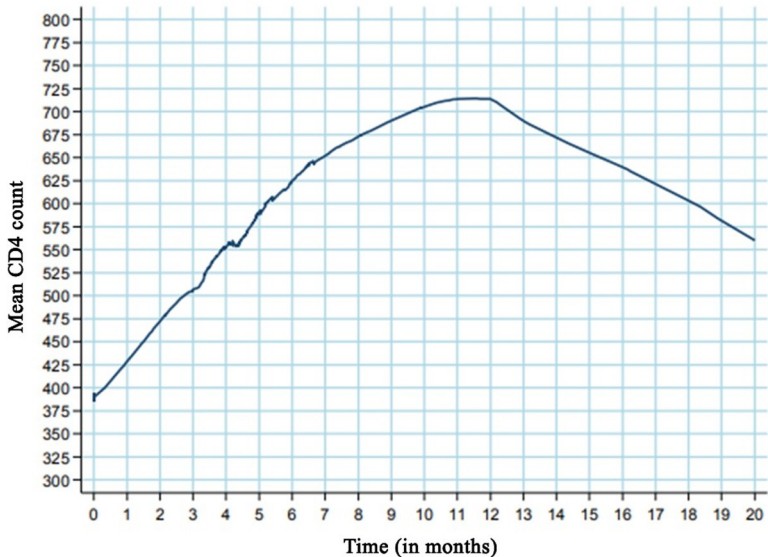

**Fig 2. Mean CD4 count from the time antiretroviral therapy was initiated in individuals after the first antiretroviral regimen containing dolutegravir.**

residents in the Northeast of Brazil, results similar to our findings [11]. Todd et al., in Northern Ireland, analysed treatment-naïve individuals on DTG with a 94% response with an undetectable viral load as early as 12 weeks, indicating a high early virologic efficacy [10]. Overall, the results reported in the present study are consistent with the results of the Single (2013), Flamingo (2014) and Todd et al. (2017) studies, which demonstrated high virologic suppression during the first weeks of treatment [7,9,10].

Dolutegravir has a high genetic barrier to mutations and, ever since the pre-clinical studies, has demonstrated a high potency against HIV. Its pharmacological profile is interesting, since it allows a low daily dose of 50 mg, with few reports of adverse events, which favours better adherence and consequently, high response rates [13,14].

In the analysis of factors associated with virologic response, we observed that individuals who maintained a detectable viral load up until week 24, had presented with a higher viral load at the beginning of treatment. However, these individuals presented a significant reduction in the number of viral RNA copies during the follow-up period, a median viral load at the beginning of ART of 334,000 copies was reduced to a median of 193 copies at week 24, thereby indicating that it would likely evolve to undetectability over a longer observation period. This finding corroborates the study by Todd et al., which reported that individuals with a higher baseline viral load needed more time to achieve undetectability [10]. Similarly, the Flamingo study, with a follow-up period of up to 96 weeks, reported a progressive reduction in the viral load even in individuals with basal levels of above 100,000 copies/mm$^3$ [7].

In the tolerability assessment, in our study, 10.4% of individuals initiating an antiretroviral regimen containing dolutegravir reported adverse events, classified as mild and moderate, data that coincides with other studies already published [7,9,10], thus reducing the possibility of discontinuing the regimen for this reason. Gastrointestinal symptoms were the most frequent (91%), and diarrhea was the most present in 43.4% of individuals.

Neuropsychiatric events were reported by 47.8% of individuals, higher than findings of a recent cohort, which was 25% [10]. Insomnia was present in 13% of this group, similar to that previously published in a controlled trial, the Single study, with a frequency of 15% with mild

insomnia. The insomnia rate was higher than in previous studies of dolutegravir [8,9]. In a recent cohort in Northern Ireland, 4% of the individuals presented a higher rate of insomnia [10]. Neuropsychiatric events have proven to be significant elements with regard to regimen change.

In this cohort, there were no serious adverse events and the reported events were insufficient for a change in the therapeutic regimen, as reported in the Single study [7], where the discontinuity rates when comparing DTG to Efavirenz were 2% versus 10%. The Flamingo study [9], which compared DTG with darunavir/ritonavir in treatment-naive subjects obtained a 2% discontinuity rate when using DTG and 4% for the other treatment.

The present study presents methodological limitations, the main is the period of follow-up of 24 weeks, that may have underestimated the effect of DTG regime, but with our results was possible to show that the proportion of virologic response corroborated with the literature findings, even with longer follow-up times. Others limitations are the use of secondary data, a nonuniformity in the period collecting the viral load and CD4 count test results. As well as patient losses in the sample, through lack of data. This was because the cohort did not interfere in the routine of the service, and followed the conduct of the medical assistant. In order to minimize this limitation, a number of searches were performed in the Laboratory Test Control System and medical records before we finally considered loss of patient follow-up due to lack of viral load information [10].

## Conclusions

The present study has provided clinically relevant data on the use of dolutegravir in first-line regimens in clinical practice outside the controlled environment of randomized trials. Dolutegravir has proved to be a drug with a high virologic response and good tolerability, with a low frequency of adverse events, which were mild and moderate. Thus, an early, sustained virologic response is critical for immune recovery by improving clinical outcomes, reducing morbidity and mortality, as well as contributing to a reduction in infectivity and in controlling the epidemic.

## Supporting information

**S1 Data.**
(RAR)

## Author Contributions

**Conceptualization:** Analú Correa, Ricardo Arraes de Alencar Ximenes, Ulisses Ramos Montarroyos.

**Data curation:** Analú Correa, Polyana Monteiro, Fernanda Calixto, Ulisses Ramos Montarroyos.

**Formal analysis:** Analú Correa, Ricardo Arraes de Alencar Ximenes, Ulisses Ramos Montarroyos.

**Funding acquisition:** Ulisses Ramos Montarroyos.

**Investigation:** Analú Correa, Polyana Monteiro, Fernanda Calixto, Joanna d'Arc Lyra Batista, Ricardo Arraes de Alencar Ximenes, Ulisses Ramos Montarroyos.

**Methodology:** Analú Correa, Polyana Monteiro, Fernanda Calixto, Joanna d'Arc Lyra Batista, Ricardo Arraes de Alencar Ximenes, Ulisses Ramos Montarroyos.

**Project administration:** Ricardo Arraes de Alencar Ximenes, Ulisses Ramos Montarroyos.

**Resources:** Ulisses Ramos Montarroyos.

**Supervision:** Ricardo Arraes de Alencar Ximenes, Ulisses Ramos Montarroyos.

**Writing – original draft:** Analú Correa, Polyana Monteiro, Fernanda Calixto, Joanna d'Arc Lyra Batista, Ricardo Arraes de Alencar Ximenes, Ulisses Ramos Montarroyos.

**Writing – review & editing:** Analú Correa, Polyana Monteiro, Fernanda Calixto, Joanna d'Arc Lyra Batista, Ricardo Arraes de Alencar Ximenes, Ulisses Ramos Montarroyos.

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
