## [Decision Letter · Decision Letter 0]

13 Jan 2020

PONE-D-19-33515

Dolutegravir: virologic response and tolerability of initial antiretroviral regimens for adults living with HIV

PLOS ONE

Dear Dr Montarroyos,

Thank you for submitting your manuscript to PLoS ONE. We have received comments from three experts in the field. Unfortunately, two of them (reviewers no. 1 and 3) consider that your paper should be rejected or needs major revision. 

Although detailed critiques are given below, according to reviewers, major limitations of your study are:

1) Conclusions of your paper are not clear. Your manuscript should specify in a clear manner its major contributions to our current knowledge on dolutegravir therapy.

2) Relevant references are lacking and proper discussion of your findings in relation to other studies is missing, It is particularly important to cite published papers on dolutegravir therapy (particularly those coming from Brazil) and clarify how conclusions of those reports differ from yours.

3) The methods section should be expanded to clarify the number and type of samples (patients) analyzed and details on how the analysis was performed,

4) The manuscript needs extensive editing. English usage and grammar should be largely improved.

In summary, we think that your manuscript needs MAJOR REVISION and you need to convince the two more critical reviewers in order to get your manuscript accepted for publication. If you are prepared to undertake the work required, I would be pleased to reconsider my decision.

We would appreciate receiving your revised manuscript by Feb 27 2020 11:59PM. To enhance the reproducibility of your results, we recommend that if applicable you deposit your laboratory protocols in protocols.io, where a protocol can be assigned its own identifier (DOI) such that it can be cited independently in the future. For instructions see: http://journals.plos.org/plosone/s/submission-guidelines#loc-laboratory-protocols

We look forward to receiving your revised manuscript.

Kind regards,

Luis Menéndez-Arias, Ph. D.

Academic Editor

PLOS ONE

Journal Requirements:

2. Please describe all virological measurements in the methods section.

3. In your Data Availability statement, you have not specified 'The data underlying the results presented in the study are available fromulisses.montarroyos@upe.br', but this can not be found.

PLOS defines a study's minimal data set as the underlying data used to reach the conclusions drawn in the manuscript and any additional data required to replicate the reported study findings in their entirety. All PLOS journals require that the minimal data set be made fully available. For more information about our data policy, please see http://journals.plos.org/plosone/s/data-availability.

Reviewers' comments:

Reviewer's Responses to Questions

**Comments to the Author**

1. Is the manuscript technically sound, and do the data support the conclusions?

Reviewer #1: No

Reviewer #2: Yes

Reviewer #3: Partly

2. Has the statistical analysis been performed appropriately and rigorously? 

Reviewer #1: No

Reviewer #2: Yes

Reviewer #3: I Don't Know

3. Have the authors made all data underlying the findings in their manuscript fully available?

Reviewer #1: No

Reviewer #2: Yes

Reviewer #3: No

4. Is the manuscript presented in an intelligible fashion and written in standard English?

Reviewer #1: Yes

Reviewer #2: Yes

Reviewer #3: Yes

5. Review Comments to the Author

Reviewer #1: Correa et al describe their experiences with DTG containing regimens in a patient cohort from Brazil. As highlighted by the authors, the data on efficacy and safety of DTG, also in real life settings, has been extensively studied. This limits the novelty of this manuscript apart from that it is from Brazil, a country where like in the main studies on DTG subtype B is dominant and where DTG has been introduced in Jan 2017. My main comment therefore is that the authors fail to make clear to me what their study and research adds to the already existing clinical information on DTG, both regarding wk24 data and tolerability. Without that, it would add little to the scientific community as low-level evidence retrospective analysis of a single center experience unfortunately.

Other major comments:

The authors state that no real life data has been published. However, publications on this do exist. For example, there is a recent publication from Brasil which highlights the high potency of DTG in an analysis of 112243 patients who were prescribed DTG or one of to other regimens (Pascom AR JIAS 2019). Also, another study from Brazil on 107647 patients (including 10.5% on DTG) showed a very favorable suppression rate on DTG which was also comparable to the suppression rate in this study (89.1% vs 90.5%). It is unclear to me how the authors can substantiate their statement or how their study differs from these studies.

Although it seems to be a retrospective analysis, the authors should state whether it was prospective or retrospective in their methods section.

The in- and exclusion criteria, data collection and the primary outcomes are not (well) defined.

The main reason to do the study was to 'aimed to estimate the frequency of individuals

with a virologic response by week 24 of antiretroviral treatment and to describe the

adverse events of the regimen containing dolutegravir'. In that perspective it is a main limitation that only from a approx 2/3 of total patients had lab data available. From the text it is unclear why the 93 patients were excluded (it reads '93 individuals provided had laboratory data'?)

Open door, but 24 weeks of follow up is too short to make any meaningful conclusions in my perspective for a real life data study and this should be acknowledged in a better way than it is currently presented. Longer follow up is required. To present the limitations of the present study as 'some methodological limitations' is an understatement in my perspective and the authors should evaluate that more thoroughly.

Minor comments:

What do the authors mean by 'bidirectional cohort study'?

Reviewer #2: The manuscript by Correa A et al. is well written and structured. The work describes a real-life experience in the use of dolutegravir in a cohort of HIV infected patients.

There are several data in literature describing the use of dolutegravir in clinical practice: I think that the Authors should improve thir discussion comparing the data presented with these other studies.

Moreover, it should be clarified if the adverse events during the treatment with dolutegravir led to the discontinuation of the regimen.

Reviewer #3: The present manuscript evaluated the efficacy and safety of a first line antiretroviral therapy based on dolutegravir. The authors concluded that dolutegravir based combination in first line therapy is a well tolerated and effective drug in the first 24 week follow-up. However, several aspects should be examined before consider it for publication.

In more detail:

Major comments:

1. Patient population:

A number of different patients have been used to assess toxicity and efficacy. This difference makes it difficult to interpret the results.

- In the results section the authors stated that “Of the 562 individuals taking dolutegravir registered on SICLOM, we identified 230 who initiated the first antiretroviral regimen containing dolutegravir” However, only 222 were included in the tolerability analysis and 137 for the virological response.

I suggest unifying the population for a better understanding of the results.

- Are there any baseline difference between the population included for tolerability and virological response?

- How many patients included in the virological study had toxicity events?

- What are the efficacy and safety data for the same population studied (f.e: 137)?

2. Immunologic recover:

- Although the baseline value of CD4 of the study population is high, 392, it is noteworthy that the average recovery of CD4 at 24-36 weeks of follow-up is only 50 cells. Given that the virological efficacy is high, what is the explanation for this relatively low immune response?

- The information recorded in figure 2 does not reflect the poor CD4 recovery mentioned along the test.

- Figure 2 represents mean CD4 counts:

o In what population?

o There is a significant increase of CD4 counts from 390 up to 715 at 12 months. However, after that there is a decrease from 715 to 550. What is the explanation for that?

Please clarify the information recorded in figure 2.

3. Virologic failure:

- For the virological analysis only 137 patients has been considered. Among this group the efficacy seems to be similar to that presented in other studies. However, have you been observed any difference when stratified patients according their baseline viral load (<100,000 vs >100,000 copies/mL)?

- There is no resistance analysis for the 15 patient with virological failure. Could you provide any data of resistance?

Minor comments:

- Page 5 line 90: “93 individuals provided had laboratory data….” You men 93 individual provided had not….?

- Page 6 line 100: “the median time using ART was 13.2 months.” If the median time on dolutegravir was 13 months why was the virologic analysis only performed at week 24?

- Table 1. Please specify if data recorded in this table correspond to 230, 222 or 137 patients.

- Table 2. Percentages of adverse events are referred to the 23 individuals who presented any event. This is confusing since it gives the impression that the adverse events rates were very high.

- Page 10 line 173. “….. of ART of 334,000 copies was reduced to a median of 193….” This information s not recorded in the result section.

6. PLOS authors have the option to publish the peer review history of their article (what does this mean?). If published, this will include your full peer review and any attached files.

Reviewer #1: No

Reviewer #2: No

Reviewer #3: No

---

## [Author Response · Author response to Decision Letter 0]

19 Jun 2020

Reviewer's Responses to Questions

Reviewer #1: Correa et al describe their experiences with DTG containing regimens in a patient cohort from Brazil. As highlighted by the authors, the data on efficacy and safety of DTG, also in real life settings, has been extensively studied. This limits the novelty of this manuscript apart from that it is from Brazil, a country where like in the main studies on DTG subtype B is dominant and where DTG has been introduced in Jan 2017. My main comment therefore is that the authors fail to make clear to me what their study and research adds to the already existing clinical information on DTG, both regarding wk24 data and tolerability. Without that, it would add little to the scientific community as lowlevel evidence retrospective analysis of a single center experience unfortunately.

Other major comments: The authors state that no real life data has been published. However, publications on this do exist. For example, there is a recent publication from Brasil which highlights the high potency of DTG in an analysis of 112243 patients who were prescribed DTG or one of to other regimens (Pascom AR JIAS 2019). Also, another study from Brazil on 107647 patients (including 10.5% on DTG) showed a very favorable suppression rate on DTG which was also comparable to the suppression rate in this study (89.1% vs 90.5%). It is unclear to me how the authors can substantiate their statement or how their study differs from these studies.

Reply from authors: Brazil have differents social and economic aspects between the regions of country and within of each region. These aspects affected the treatment response, treatment adherence, health support of patient by local heath system, because this is important that local studies to be carried and the evidence compared with the national reality. 

Although it seems to be a retrospective analysis, the authors should state whether it was prospective or retrospective in their methods section.

Reply from authors: The study was a retrospective cohort. Corrected in method.

The in- and exclusion criteria, data collection and the primary outcomes are not (well) defined.

Reply from authors: Included the exclusion criteria and the primary outcome in the article text and reformulated the description of data collection. 

The main reason to do the study was to 'aimed to estimate the frequency of individuals with a virologic response by week 24 of antiretroviral treatment and to describe the adverse events of the regimen containing dolutegravir'. In that perspective it is a main limitation that only from a approx 2/3 of total patients had lab data available. From the text it is unclear why the 93 patients were excluded (it reads '93 individuals provided had laboratory data'?)

Reply from authors: For standardization purposes, all viral load and CD4 records of the study participants were collected following the hospital routine. Due to this, is expected not all patients had viral load and CD4 cell count measurements consistently. Of 230 patients, 93 were exclude for don’t had at least two available HIV viral load measurements collected. The data of adverse events was collected from patient records, data that just 8 patients don’t had informed in the medical records. 

Open door, but 24 weeks of follow up is too short to make any meaningful conclusions in my perspective for a real life data study and this should be acknowledged in a better way than it is currently presented. Longer follow up is required. To present the limitations of the present study as 'some methodological limitations' is an understatement in my perspective and the authors should evaluate that more thoroughly.

Reply from authors: We consider the main methodological limitations the period of follow-up of 24 weeks, that may have underestimated the effect of DTG regime, but was possible to show that the proportion of virologic response corroborated with the literature findings, although the observation period was short.

Minor comments: What do the authors mean by 'bidirectional cohort study'?

Reply from authors: The study was a retrospective cohort. Corrected in method.

Reviewer #2: The manuscript by Correa A et al. is well written and structured. The work describes a real-life experience in the use of dolutegravir in a cohort of HIV infected patients. There are several data in literature describing the use of dolutegravir in clinical practice: I think that the Authors should improve their discussion comparing the data presented with these other studies. Moreover, it should be clarified if the adverse events during the treatment with dolutegravir led to the discontinuation of the regimen.

Reply from authors: In our findings the discontinuation of the regimen occurred in just 9 of 222 participants (4.1%). The reason on 6 patients was the tuberculosis treatment, disease with high prevalence in the studied region, and the other 3 cases was due to pregnancy, an individual with a psychiatric disease prior to initiating ART, and one for reasons unregistered in the medical records.

Reviewer #3: The present manuscript evaluated the efficacy and safety of a first line antiretroviral therapy based on dolutegravir. The authors concluded that dolutegravir based combination in first line therapy is a well tolerated and effective drug in the first 24 week follow-up. However, several aspects should be examined before consider it for publication. In more detail: 

Major comments: 1. Patient population: A number of different patients have been used to assess toxicity and efficacy. This difference makes it difficult to interpret the results. - In the results section the authors stated that “Of the 562 individuals taking dolutegravir registered on SICLOM, we identified 230 who initiated the first antiretroviral regimen containing dolutegravir” However, only 222 were included in the tolerability analysis and 137 for the virological response. I suggest unifying the population for a better understanding of the results.

Reply from authors: For standardization purposes, all viral load and CD4 records of the study participants were collected following the hospital routine. Due to this, is expected not all patients had viral load and CD4 cell count measurements consistently. Of 230 patients, 93 were exclude for don’t had at least two available HIV viral load measurements collected. The data of adverse events was collected from patient records, data that just 8 patients don’t had informed in the medical records. We think to analysis the adverse event with the 137 patients but we considerate that the discrepancy not produce selection bias. 

 - Are there any baseline difference between the population included for tolerability and virological response? 

Reply from authors: Comparing the characteristic of patients, the frequency of males there was statistic difference. We believe that this difference not modify the estimate of virological response, the association with virological response not was significant in our study (p = 0.623 – table 3). 

- How many patients included in the virological study had toxicity events? What are the efficacy and safety data for the same population studied (f.e: 137)? 

Reply from authors: Of the 137 individuals included in the virological study, 21 had adverse event. 

2. Immunologic recover:

- Although the baseline value of CD4 of the study population is high, 392, it is noteworthy that the average recovery of CD4 at 24-36 weeks of follow-up is only 50 cells. Given that the virological efficacy is high, what is the explanation for this relatively low immune response? 

Reply from authors: Our results show an increase mean of 50 cells each month. We did correction in the text making clear that is a monthly growth mean rate. Considering 24 weeks the mean of CD4 count of population was 625 cell and in the 36 weeks was 700 cells, approximately. We conclude that the immune response was satisfactory. 

- The information recorded in figure 2 does not reflect the poor CD4 recovery mentioned along the test. - Figure 2 represents mean CD4 counts: 

o In what population? 

Reply from authors: individuals after the first antiretroviral regimen containing Dolutegravir.

o There is a significant increase of CD4 counts from 390 up to 715 at 12 months. However, after that there is a decrease from 715 to 550. What is the explanation for that? Please clarify the information recorded in figure 2.

Reply from authors: Although CD4+ cell count is an important test for monitoring antiretroviral therapy effectiveness and guiding opportunistic infection prophylaxis, values may vary from one measurement to another, fluctuating with diurnal variation and ongoing infections, especially with higher CD4+ counts. Once CD4+ cell count is good, it requires less frequent monitoring and practioners should look at overall measurements. In this regard, the viral load is the most important parameter in treatment monitoring. 

3. Virologic failure: - For the virological analysis only 137 patients has been considered. Among this group the efficacy seems to be similar to that presented in other studies. However, have you been observed any difference when stratified patients according their baseline viral load (<100,000 vs >100,000 copies/mL)? - There is no resistance analysis for the 15 patient with virological failure. Could you provide any data of resistance? 

Reply from authors: Among the 15 patients with virological failure, six achieved virological control beyond 24 weeeks, it shoul be noted that all of them had baseline viral load >100,000 copies/mL. Despite the fact that antiretroviral regimens with dolutegravir facilitate adherence they do not guarantee it totally, adherence is multifactorial and varies within individuals over time. Even though convenience and tolerability of ART have improved, suboptimal adherence remains common (Bangsberg DR, Kroetz DL, Deeks SG. Adherence-resistance relationships to combination HIV antiretroviral therapy. Curr HIV/AIDS Rep. 2007 May;4(2):65-72; Dorward J, Lessells R, Drain PK et al. Dolutegravir for first-line antiretroviral therapy in low-income and middle-income countries: uncertainties and opportunities for implementation and research. Lancet HIV. 2018 Jul; 5(7): e400–e404). We found that nine patients who did not achieve virological control had suboptimal adherence with irregular follow-up visits . Two of them had genotypic assays with no evidence of resistance mutations and one patient had no genotypic data since viral load was low (<1000 copies/mL).

Minor comments: 

- Page 5 line 90: “93 individuals provided had laboratory data….” You men 93 individual provided had not….? 

Reply from authors: Done 

- Page 6 line 100: “the median time using ART was 13.2 months.” If the median time on dolutegravir was 13 months why was the virologic analysis only performed at week 24? 

Reply from authors: Some individuals we considerate the CD4 count and viral load after the investigation period with objective to describe the CD4 cells and load viral behavior in the maximum time. 

- Table 1. Please specify if data recorded in this table correspond to 230, 222 or 137 patients. 

Reply from authors: Done 

Table 2. Percentages of adverse events are referred to the 23 individuals who presented any event. This is confusing since it gives the impression that the adverse events rates were very high. 

Reply from authors: Modified in the text of article 

Page 10 line 173. “….. of ART of 334,000 copies was reduced to a median of 193….” This information s not recorded in the result section. 

Reply from authors: Included in the text of article – “The median viral load of patients at the beginning of ART was 334,000 copies, after 24 weeks decrease to 193 copies”.

---

## [Decision Letter · Decision Letter 1]

10 Aug 2020

Dolutegravir: virologic response and tolerability of initial antiretroviral regimens for adults living with HIV

PONE-D-19-33515R1

Dear Dr. Montarroyos,

We’re pleased to inform you that your manuscript has been judged scientifically suitable for publication and will be formally accepted for publication once it meets all outstanding technical requirements.

Kind regards,

Luis Menéndez-Arias, Ph. D.

Academic Editor

PLOS ONE

Additional Editor Comments (optional):

Reviewers' comments:

Reviewer's Responses to Questions

**Comments to the Author**

1. If the authors have adequately addressed your comments raised in a previous round of review and you feel that this manuscript is now acceptable for publication, you may indicate that here to bypass the “Comments to the Author” section, enter your conflict of interest statement in the “Confidential to Editor” section, and submit your "Accept" recommendation.

Reviewer #1: All comments have been addressed

2. Is the manuscript technically sound, and do the data support the conclusions?

Reviewer #1: Yes

3. Has the statistical analysis been performed appropriately and rigorously? 

Reviewer #1: Yes

4. Have the authors made all data underlying the findings in their manuscript fully available?

Reviewer #1: Yes

5. Is the manuscript presented in an intelligible fashion and written in standard English?

Reviewer #1: Yes

6. Review Comments to the Author

Reviewer #1: I have no further comments for the authors on this submitted manuscript. My questions have been addressed.

7. PLOS authors have the option to publish the peer review history of their article (what does this mean?). If published, this will include your full peer review and any attached files.

Reviewer #1: **Yes: **Dr. Casper Rokx

---

## [Editor Report · Acceptance letter]

20 Aug 2020

PONE-D-19-33515R1 

Dolutegravir: virologic response and tolerability of initial antiretroviral regimens for adults living with HIV 

Dear Dr. Montarroyos:

I'm pleased to inform you that your manuscript has been deemed suitable for publication in PLOS ONE. Congratulations! Your manuscript is now with our production department. 

Kind regards, 

on behalf of

Dr. Luis Menéndez-Arias 

Academic Editor

PLOS ONE